# Endemic *Veronica saturejoides* Vis. ssp. *saturejoides*–Chemical Composition and Antioxidant Activity of Free Volatile Compounds

**DOI:** 10.3390/plants9121646

**Published:** 2020-11-25

**Authors:** Marija Nazlić, Dario Kremer, Renata Jurišić Grubešić, Barbara Soldo, Elma Vuko, Edith Stabentheiner, Dalibor Ballian, Faruk Bogunić, Valerija Dunkić

**Affiliations:** 1Faculty of Science, University of Split, Ruđera Boškovića 33, HR-21000 Split, Croatia; marija.nazlic@pmfst.hr (M.N.); barbara@pmfst.hr (B.S.); elma@pmfst.hr (E.V.); 2Faculty of Pharmacy and Biochemistry, University of Zagreb, A. Kovačića 1, HR-10000 Zagreb, Croatia; dkremer@pharma.hr (D.K.); rjurisic@pharma.hr (R.J.G.); 3Institute of Biology, Karl-Franzens University, Schubertstrasse 51, A-8010 Graz, Austria; edith.stabentheiner@uni-graz.at; 4Faculty of Forestry, University of Sarajevo, Zagrebačka 20, BIH-71000 Sarajevo, Bosnia and Herzegovina; balliandalibor9@gmail.com (D.B.); faruk_bogunic@yahoo.com (F.B.); 5Slovenian Forestry Institute, Večna pot 2, SI-1000 Ljubljana, Slovenia

**Keywords:** antioxidant activity, DPPH, GC, GC-MS, glandular trichomes, ORAC, polyphenols, *Veronica saturejoides*, volatile compounds

## Abstract

Chemical profile and antioxidant activity of the species *Veronica saturejoides* Vis. ssp. *saturejoides* (Plantaginaceae)—which is endemic to Croatia, Bosnia and Herzegovina and Montenegro —were investigated. Volatile compounds produced by glandular trichomes (composed of one stalk cell and two elliptically formed head cells according to scanning electron microscope investigation) were isolated from the plants collected in two locations. Additionally, as a part of specialized metabolites, total polyphenols, total tannins, total flavonoids and total phenolic acids were determined spectrophotometrically. In the lipophilic volatile fractions-essential oils, the most abundant compounds identified were hexahydrofarnesyl acetone, caryophyllene oxide and hexadecanoic acid. In total, the class of oxygenated sesquiterpenes and the group of fatty aldehydes, acids and alcoholic compounds dominated in the essential oils. In the hydrophilic volatile fractions-hydrosols, the most abundant compounds identified were *trans-p-*mentha-1(7),8-dien-2-ol, *allo*-aromadendrene and *(E)-*caryophyllene. A group of oxygenated monoterpenes and the sesquiterpene hydrocarbons dominated in the hydrosols. Antioxidant activity of essential oils and hydrosols was tested with two methods: 2,2′-diphenyl-1-picrylhydrazyl (DPPH) and oxygen radical absorbance capacity (ORAC). Essential oils showed higher antioxidant activity than hydrosols and showed similar antioxidant activity to *Rosmarinus officinalis* essential oil. Obtained results demonstrate that this genus is a potential source of volatiles with antioxidant activity.

## 1. Introduction

The genus *Veronica* L. is the largest within the order Lamiales (family Plantaginaceae) with about 450 species. The extreme variability in morphology and the very good adaptation to the different living conditions of this genus has allowed it to be widely distributed on a wide range of habitats, from aquatic, swamp and forest habitats to rocks, cracks, fields and ruderal habitats [1]. Most representatives grow in areas with a Mediterranean climate [2]. Between 30 and 40 species of the genus *Veronica* have been described in Croatia [3], but the species studied in this research is the only endemic species. *Veronica saturejoides* Vis. ssp. *saturejoides* grows on the rocks in the Dinaric Mountains in Croatia, Bosnia and Herzegovina and Montenegro. There are two other subspecies of this plant, one from Albania, *V. saturejoides* ssp. *munellensis*, and one from Bulgaria, *V. saturejoides* ssp. *kellereri* [4]. *V. saturejoides* Vis. ssp. *saturejoides* is a perennial crawling plant, which grows 10-30 cm long and has elongated, somewhat lignified roots. It has a hairy stem which is woody at the base. The leaves are simple, opposite each other, 6–9 mm long, have an integrated edge and are not or only slightly hairy on the leaves. The blue flowers are androgynous and zygomorphic [5].

Genus *Veronica* has been extensively phylogenetically investigated and the relationship between the distribution of iridoid glycosides and phylogeny has been reported [6,7]. Flavonoid and phenolic compounds have also been extensively studied, probably because of their importance for the biological activity of plants. These compounds are known to have anti-allergenic, antiviral, anti-inflammatory, cardioprotective and vasodilator properties and, above all, antioxidant and radical scavenging potential [8]. Species of the genus *Veronica* are used in traditional medicine for the treatment of various diseases, including influenza, respiratory diseases and cancer, and as diuretics [9] because of their natural richness in phenolic compounds and iridoids. In recent years, some research has shown that species from this genus could also be used to treat mental disorders [10] and some types of diabetes [11].

Free volatiles of the genus *Veronica* are not very well studied. To our knowledge, *V. saturejoides* ssp. *saturejoides* has not yet been studied, which encouraged our team to study free volatile components of this endemic species, especially in terms of comparing volatile components from essential oil (EO) and from water residues, and to test the antioxidant activity of these extracts, since this species grows under extreme environmental conditions and we assumed that it develops volatile components that could have antioxidant activity. In this research, volatile substances are extracted from glandular trichomes, which are later shown in the pictures. Free volatiles were isolated in lipophilic and water fraction. Water fractions or hydrosols are condensed water vapors containing dissolved molecules of EOs and more water-soluble (polar) volatile compounds [12]. Due to the different solubility of the volatile compounds in water, the overall composition and thus the biological activity of the hydrosol differs from lipophilic fraction or essential oil. Hydrosols are often discarded after the EO extraction process. However, research has shown that these waste products are rich in biologically active substances [12]. Hydrosols from various plants are becoming increasingly important in the food industry, the cosmetics industry, the application of plant pesticides, traditional pharmaceuticals, in aromatherapy as part of complex formulations and as independent products; therefore, their potential use should be further investigated [13].

We now know that plants are exposed to many stresses in their environment. Due to their sedentary lifestyles, they develop specialized metabolites in response to these extreme weather conditions [14] (for this plant, drought and high levels of sunlight due to growing on mountain peaks) and biological stress such as pathogen infection. Specialized plant metabolites have all kinds of biological activities that can be used in medicine, pharmacy and food preservation, as microorganisms become increasingly resistant to synthetic compounds. These synthetic compounds (e.g., BHA-butylated hydroxyanisole) can also be carcinogenic when used in canned foods [9], therefore the search for safe natural food preservatives goes on and this research contributes to the study of natural products as potential antioxidants in food preservations. What is more, it is basic research on the bioactive compounds of *Veronica* genus.

## 2. Results and Discussion

### 2.1. Gas Chromatography and Mass Spectrometry (GC-MS) Analysis of the Free Volatile Compounds from Essential Oils and Hydrosols

Two samples of EO and hydrosol obtained from *V. saturejoides* ssp. *saturejoides* were analyzed. The results are presented in Table 1. The yield of EO for two samples, Prenj (PS) and Kamešnica (KS), was 0.07% and 0.03% respectively. This endemic species is rich in volatile compounds that have been studied in oils and hydrosols. The compounds are listed in the order of their elution from the column (Table 1). In total, the class of oxygenated sesquiterpenes (PS 54.25% and KS 36.46%) and the group of fatty aldehydes, acids and alcoholic compounds (PS 17.4% and KS 43.85%) dominate in the oils, if one considers the results obtained. Within the hydrosol, the oxygenated monoterpenes (PS 31.75% and KS 40%) and the sesquiterpene hydrocarbons (PS 36.46% and KS 34.53%) dominate (Table 1, Figure 1).

In the PS essential oil, 25 compounds were identified, which account for 92.47% of the total oil. The most abundant compound was hexahydrofarnesyl acetone (30.13%). A total of 21 compounds were identified in KS essential oil, representing 95.38% of the total oil, with the most abundant compound being hexadecanoic acid (37.31%). Table 1 shows that the number of compounds detected and identified in hydrosol was lower than in EO. This is logical since non-polar compounds are less soluble in water. In the PS hydrosol, we identified 17 compounds, representing 92.29% of the total hydrosol; in the KS hydrosol, we identified 16 compounds, representing 92.43% of the total hydrosol. In the hydrosol samples, the most abundant compound was *trans-p-*mentha-1(7),8-dien-2-ol, which represented 31.75% and 36.63% of the total hydrosol in the PS and KS, respectively. These results are consistent with the fact that hydrosols contain dissolved molecules of EO, considering that more than half of the compounds that we identified in hydrosol can also be found in the essential oil [12].

Hexahydrofarnesyl acetone (phytone) and hexadecenoic acid (palmitic acid) are important components in both samples of the EOs analyzed in this study. According to the literature, phytone shows strong antimicrobial activity and broad-spectrum inhibition against various fungal strains [15]. Palmitic acid has antioxidant, nematidical, pesticide, antiandrogenic, hemolytic and 5-alpha reductase activities [16]. Palmitic acid was also found in the EO of the species *Veronica thymoides* P. H. Davis subsp. *pseudocinerea* M. A. Fischer, where it accounted for 5.4% of the total EO [17].

The presence of hydrocarbons is significant, especially in the oil PS where pentacosan dominates with 6.28%. Sesquiterpenes (E)-caryophyllene and caryophyllene oxide were identified in both oil samples. Caryophyllene oxide is particularly present in the PS oil with 20.25%, while it is almost ten times less present in KS oil (2.34%). Sesquiterpenes are also significantly represented and dominate in the analyzed hydrosols. The proportion of identified (E)-caryophyllene in the hydrosol samples (24.52% and 12.35%) is significantly higher than in the oil samples, followed by allo-aromadendrene (8.13% and 11.53%) and germacrene D (2.56% and 4.67%), while caryophyllene oxide is least present in the hydrosols samples (Table 1). Volatile components rich in sesquiterpenes are known to have antifungal, antimicrobial, anticancer and antioxidant properties [18,19,20,21]. In addition to these sesquiterpenes and *trans-p-*mentha*-*1(7),8-dien-2-ol in the analyzed hydrosol samples, methyl eugenol was significantly present with 13.35% in the PS hydrosol and 11.92% in the KS hydrosol. Phenylpropanoids such as methyl eugenol and identified Z-methyl isoeugenol (1.25% and 4.16%) occur in plants under stress conditions, such as ultraviolet radiation and pathogen attack [22].

In the previously investigated *Veronica spicata* L., phytol was the dominant compound (21.13%) in the total EO [9]. In this research phytol was only present in the PS essential oil (Table 1) (2.82%). A further comparison of the compounds found in *V. spicata* L. and *V. saturejoides* shows that nine compounds were found in both plant species: (E)-caryophyllene, spathulenol, caryophyllene oxide, γ- eudesmol, phytol, docosane, tricosane, tetracosane and pentacosane. Li (2002) [23] identified essential oil components of *V. linariifolia* Pall. ex Link and found that the main components were cyclohexene (25.83%), β-pinene (11.61%), 1S-α-pinene (10.65%), β-phellandrene (10.49%), β-myrcene (10.42%), and germacrene D (4.99%). If we compare these results with ours, we can conclude that EO of *V. saturejoides* is richer in oxygenated sesquiterpenes and *V. linariifolia* in sesquiterpene hydrocarbons. Çelik et al. investigated EOs extracted from *Veronica* sp. and found that the main components were mainly linalool (4.18%) and carvacrol (7.28%) [24]. Research on free volatile compounds of the genus *Veronica* is very scarce, therefore it is important that our research is supported by the micromorphology of trichomes as a visible evidence of the place of synthesis of free volatile compounds.

### 2.2. Micromorphological Traits

In general, micromorphological investigations of *Veronica* species are rare, so we have decided to conduct the micromorphological research on this plant to find site of essential oil production. On the stems, leaves, and the calyces of *V. saturejoides* non-glandular and glandular trichomes could be observed. According to scanning electron microscope (SEM) investigation, non-glandular trichomes (Figure 2a,b,g,h) were unbranched, bi-cellular to multicellular, uniseriate, and folded at different levels. They could be noted as attenuate trichomes [17]. The length of these trichomes varied from very short to long trichomes (Figure 2g,h). These trichomes protect the plant from water loss and maintain the positive microclimate. The surfaces of these trichomes showed a warty appearance due to the occurrence of cuticular micropapillae (Figure 2g,h). Leaves and flowers were sparsely covered by non-glandular trichomes (Figure 2a–f), while the stem was characterized by a relatively dense indumentum of these trichomes (Figure 2g,h). The existence of non-glandular trichomes on flower parts of *Veronica* species was mentioned 100 years ago by Kurer [26]. Additionally, Kraehmer and Baur described these trichomes in *V. persica* Poir [27]. The same type of non-glandular trichomes is common in many other Lamiaceae species [28,29,30].

Glandular capitate trichomes could be observed on stems, leaves, and the calyces of *V. saturejoides* ssp. *saturejoides*. These trichomes were composed of one stalk cell and two elliptically formed head cells (Figure 2e). They were not upright, and could be described as clinging to the surface. All investigated plant parts were sparsely covered by capitate trichomes. The same type of capitate trichomes was noticed in *V. beccabunga* L. [31]. Likewise, an inclined trichome type with a bicellular head was reported in *Stachys recta* L. subsp. *recta* by Vundac et al. [32]. Comparable capitate trichomes with only one elliptically formed head cell could be observed on SEM micrographs of *Marrubium vulgare* L. (Lamiaceae) in research by Haratym and Weryszko-Chmielewska [28]. Moreover, Hanlidou et al. described a similar trichome type (“short and ordinarily bent”) in *Calamintha menthifolia* Host. [33]. Kremer et al. also found an inclined trichome type with one head cell in *Micromeria croatica* (Pers.) Schott [34]. Although the yield of EO was considerably higher in PS (0.07%) than in KS (0.03%), the micrographs did not show any significant difference between the samples for the number of capitate trichomes on calyces and leaves (Table 2, Figure 2c–f). A slightly higher number of capitate trichomes was found on stems from PS (Figure 2g). The obtained difference in EO yield between samples could be due to other reasons, such as climatic conditions or possible mechanical damage of the glandular trichomes.

### 2.3. Polyphenol Analysis in Dry Plant Material

Plant polyphenols are natural biologically active compounds that can also be synthesized in the laboratory. They have been shown to be good antioxidants, anti-neurodegenerative and anticancer agents [35]. In our research, as is shown in Table 3, the highest content was found for total phenolic acids in *V. saturejoides* ssp. *saturejoides* (Kamešnica-KS; 1; A525 nm), while the yield of total flavonoids (TF) was found to be very low and the same for both investigated specimens of *V. saturejoides* ssp. *saturejoides* (Table 2). Harput et al. found that total phenolic content was 200.20 mg/g in *V. officinalis* L., 139.92 mg/g in *V. peduncularis* M. Bieb., 127.64 mg/g in *V. orientalis* Mill., and 83.15 mg/g in *V. baranetzkii* Bordz. [36]. Comparing these results with ours, it can be seen that *V. saturejoides* has a similar total phenolic content to *V. baranetzkii*. Ertas et al. found that the total phenolic content in methanolic extracts of *V. thymoides* subsp. *pseudocinerea* was 248.37 ± 3.68 mg/g and that total flavonoid content was 47.02 ± 0.21 mg/g [17].

The absorbances obtained at 505 nm refer to rosmarinic acid, and the results at 525 nm represent the chlorogenic acid content. According to the pharmacopoeial expressions (A_505nm_ × 2.5/m; A_525nm_ × 5.3/m), and taking into account specific absorbances of standard phenolic acids (rosmarinic acid $A_{1cm}^{1\%}$= 400; chlorogenic acid $A_{1cm}^{1\%}$= 180), chlorogenic acid predominates in the tested plant samples.

### 2.4. Phenolic Compounds in Hydrosols

Phenolic compounds in hydrosols were analyzed for better explanation of antioxidant activity of the hydrosols. Vanillin, cinnamic acid, and protocatechuic acid (3, 4-dihydroxybenzoic acid) were confirmed. All three compounds were found in hydrosol of plants collected in Kamešnica (KS). In hydrosols of PS (from Prenj), polyphenolic compounds were not detected (Table 4). In all analyzed samples from Kamešnica, protocatechuic acid was the most abundant compound, with an average concentration of 7.33 ± 0.35 mg L^−1^. Phenolic compounds generally have a protective role in plants and in their acclimatization to environmental conditions, so that the difference between detected phenols in PS and KS hydrosols can be attributed to exposure to different stresses (i.e., water stress). Concentrations of vanillin and cinnamic acid ranged from 0.11 mg to 0.22 mg L^−1^. The average vanillin concentration was 0.22 ± 0.01 mg L^−1^ and that of cinnamic acid was 0.12 ± 0.02 mg L^−1^. Low amounts of polyphenols were expected due to their insolubility in water. In four types of *V. spicata* plant extracts (methanol, ethyl-acetate, water at 25 °C, and water at 45 °C), cinnamic acid and vanillin were not confirmed [9]. The concentration of protocatechuic acid was in the range from 0.008 ± 0.001% (in methanol) to 0.151 ± 0.012% (water at 45 °C). In ethyl-acetate extracts, protocatechuic acid was not found [9]. Beara et al. also found significant amounts of protocatechuic acid in 70% aqueous acetone extracts of *V. teucrium* and *V. jacquinii* [37]. Stojković et al. found that protocatechuic acid was the main compound in water extracts of *V. montana* L. [38]. 

### 2.5. Antioxidant Activity

In Table 5, we can see that KS oil from the Kamešnica Mountains showed slightly higher oxygen radical absorbance capacity (ORAC) and 2,2′-diphenyl-1-picrylhydrazyl (DPPH) activity. In the polyphenolic analyses in the sample from Prenj Mountain, we could not detect any polyphenols, only in the sample from the Kamešnica Mountain, so we can conclude that this small difference in antioxidant activity could be due to this finding. This is the first report on the antioxidant activity of the free volatiles of *Veronica* species, so we cannot compare it with other results for *Veronica* species, but we can compare our results with other relevant EOs and hydrosols commonly used in food preservation, preservation, pharmacy and cosmetics. In comparison with reports on other plants, Aazza et al. [39] investigated the ORAC activity of hydrosols of several medicinal plants (*Lavandula officinalis, Origanum majorana, Rosmarinus officinalis, Salvia officinalis, Thymus vulgaris, Cinnamomum verum* and *Syzgium aromaticum*). The hydrosol KS tested in our research has higher ORAC activity than *Salvia officinalis* and similar activity to *Rosmarinus officinalis* [39]. Bentayeb et al. [40] reported antioxidant activity for 10 EOs of the plants often used as spices. ORAC antioxidant activity of *V. saturejoides* according to their reported values has similar activity as rosemary and basil EO. Viuda-Martos et al. [41] reported antioxidant activity for five plants used in Mediterranean and *V. saturejoides* showed similar DPPH activity to rosemary oil which is in accordance to other reported results [41]. Antioxidant activity for metabolites of the genus *Veronica* was mostly tested on different extracts of phenolic compounds and on iridoid glucosides. Harput et al. [42] tested the DPPH activity of methanolic extracts from fourteen different *Veronica* species and reported the highest antioxidant activity in *V. officinalis* with an IC50 value of 40.93 μg/mL [42]. Mocan et al. [43] confirmed the antioxidant activity for *V. officinalis* with TEAC method 157.99 ± 6.58 mg TE/g. Živković et al. [10] tested methanolic extracts of *V. teucrium* and *V. jacquinii* and they were stronger than previously reported by Harput et al. [42] for *V. officinalis*, with IC50 values of 28.49 ± 0.6 for *V. teucrium* and 37.63 ± 0.6 μg/mL for *V. jacquinii*, although both have lower activity than the standard (BHT-butylated hydroxytoluene and BHA-butylated hydroxyanisole) [10]. Kwak et al. reported antioxidant activity for iridoid glucosides in which the ethanolic extract showed higher activity than Trolox [44]. In our previous study, we tested the DPPH activity of various extracts of *V. spicata*, and the IC50 value showed the highest activity in methanolic extracts of flowers and leaves with values of 8.21 ± 0.06 μg/mL and 8.69 ± 0.06 μg/mL, i.e., higher than the values for standards BHT and BHA according to the values in the study by Živković et al. [10]. 

We have also tested the most abundant compound in EOs hexahydropharnesyl acetone with DPPH and ORAC methods, and it did not show any antioxidant activity, so we can assume that the antioxidant activity comes from another compound in EO, or is probably the result of synergistic work between different compounds. Synergistic activity has been demonstrated in some studies. Amorati et al. tested and compared the antioxidant activity for 5 different EOs, their hydrocarbon and oxygenated fractions and also for two compounds characteristic for these EOs, thymol and carvacrol. The results showed that in 4 out of 5 samples, the total oil has a higher antioxidant activity than its fractions, and in 3 out of 5 samples, the total oil has a higher activity than isolated single active compounds, so that we can conclude that the synergy between different compounds in the essential oil plays a crucial role in its activity [45]. From our results, we can also conclude that the activity does not originate from the most abundant compound, but from the interplay between all compounds from the EO. Our previous research showed similar results with phenolic extracts, since for *V. spicata*, it was shown that there is a negative correlation between the amount of phenolic compounds and antioxidant activity, so that it was concluded that other substances such as terpenoids and proteins may have an influence on antioxidant activity [9]. We have shown that *Veronica* species have free volatile compounds (terpenoids) and that they have antioxidant activity. Other potential biological activity such as antimicrobial, antiviral and antiproliferative should be carried out to give better insight into the full potential of using these compounds in pharmacy or food preservation.

## 3. Materials and Methods

### 3.1. Herbal Material

Randomly selected samples of wild-growing plants of *V. saturejoides* Vis. were collected at the end of the blooming period in July 2018 from two locations, one in Croatia and one in Bosnia and Herzegovina: sample 1 (Prenj sample – PS) – Prenj Mountain (Bosnia and Herzegovina; 43°43′16″ N, 18°07′03″ E; 1525 m a.s.l.; Voucher No. HFK-HR 121/2018); sample 2 (Kamešnica sample – KS) –Kamešnica Mountain (Croatia; 43°43′33″ N, 16°51′57″ E; 1568 m a.s.l.; Voucher No. HFK-HR 122/2018). Voucher specimens of herbal material were deposited in the “Fran Kušan“ herbarium (HFK-HR), Faculty of Pharmacy and Biochemistry, University of Zagreb, Croatia.

For gas chromatography (GC-FID) and gas chromatography-mass spectrometry (GC and GC-MS) analyses, samples were air dried for three weeks in a single layer in a well-ventilated room and protected from direct sunlight. Dried plant material was placed in double paper bags labeled with the sample number and stored in a dry place away from light until analysis.

For micromorphological studies of trichomes, samples from seven plants per locality were fixed in FAA (formalin/96% ethanol/acetic acid/water: 5/70/5/20). After three days, the samples were transferred to 70% ethanol (Kemika, Zagreb, Croatia) and stored in fridge until analysis.

### 3.2. GC and GC-MS Analyses

Dried aerial parts (50 g) for each location were hydrodistilled for three hours using Clevenger-type apparatus. For each sample, we collected lipophilic (extracted in the pentane part in the inner tube of the Clevenger apparatus) and hydrophilic volatile compounds (extracted in the water part in the inner tube of the Clevenger apparatus) fractions and stored in fridge until analysis. Both phases were analyzed with GC and GC-MS. GC was performed by gas chromatograph (model 3900, Varian Inc., Lake Forest, CA, USA) that is supplied with a flame ionization detector (FID), mass spectrometer (model 2100T; Varian Inc.), non-polar capillary column VF-5ms (30 m × 0.25 mm inside diameter, coating thickness 0.25 µm, Palo Alto, CA, USA) and polar capillary column CP-Wax 52 CB (30 m × 0.25 mm i.d., coating thickness 0.25 µm, Palo Alto, CA, USA). The chromatographic conditions for the analysis of lipophilic fraction (essential oils) were: FID detector temperature 300 °C, injector temperature 250 °C. The gas carrier was helium at 1 mL min^−1^. The conditions for the VF-5ms column were: temperature 60 °C isothermal for 3 min, and then increased to 246 °C at a rate of 3 °C min^−1^, and held isothermal for 25 min. Conditions for the CP Wax 52 column were: temperature 70 °C isothermal for 5 min, and then increased to 240 °C at a rate of 3 °C min^−1^ and held isothermal for 25 min. The injected volume was 2 μL and the split ratio was 1:20. The MS conditions were: ion source temperature 200 °C, ionization voltage 70 eV, mass scan range 40–350 mass units [9]. The individual peaks for all samples were identified by comparing their retention indices of n-alkanes with those of authentic samples and literature [25]. The chromatographic conditions for the analysis of hydrophilic fraction (hydrosols) were the same, however the injection was done with a headspace injection needle and there was no split ratio (splitless mode). The procedure for each hydrosol sample was as follows: 2 g of hydrosol was added in the glass bottle and closed with a metal cap with septum. The headspace needle was injected in the glass bottle closed with metal cap with septum. The glass bottle was first placed with the hydrosol sample in water at 40 °C and left there for 20 min without the needle to allow volatile compounds to evaporate from the water. The needle was then injected and left there for 20 min so that the volatile compounds could be adsorbed on the resin needle. The injection needle was then inserted into a GC inlet and left there for 20 min to ensure that all volatile compounds from the resin were resorbed into the injection liner. The MS conditions were the same as for volatile compounds from EO and the individual peaks for all samples were identified by comparing of their retention indices of n-alkanes with those of authentic samples and literature [25]. The results for all samples were measured in three independent analyses and expressed as percentage (%) of each compound in a total EO or hydrosol (Table 1). All values were calculated as the mean of three independent results with standard deviation.

### 3.3. Micromorphological Traits

For SEM investigation, stem, leaf, and calyx samples were transferred from 70% ethanol to 70% acetone and then further dehydrated (70%, 90% and 100% acetone) and subjected to critical point drying using CO_2_ as drying medium (CPD030; Bal-tec, Balzers, Liechtenstein). The samples were then sputtered with gold (Sputter Coater, AGAR) and examined under an XL30 ESEM (FEI) SEM with an acceleration voltage of 20 kV in high vacuum mode [46]. Common terminology [47] was used to describe the trichomes.

### 3.4. Phenolic Compounds in Hydrosols

The phenolic compounds of the hydrosols were separated by high- performance liquid chromatography (HPLC) on a Series 200 Perkin Elmer HPLC system (Waltham, MA, USA), equipped with a thermostated autosampler, vacuum degasser, binary pump, thermostated column section, UV/VIS detector and the TotalChrom Workstation software package (version 6.2.1, Perkin Elmer, Waltham, MA, USA) used for the analyses. Phenolic compounds were separated with the C18 column (Ultra-Aqueous C-18, 250 × 4.6 mm, 5 Å) (Restek, Bellefonte, PA, USA) and by gradient chromatography. Chromatography conditions were set according to the methodology by Jukic Spika et al. [48].

The identification of phenolic compounds in the hydrosols was performed by comparing the retention time with that of the pure standard. The quantification of the phenolic compounds was performed using the calibration curve of the standard and the results were expressed in mg of each phenolic compound per L of the sample. The standard and solvents were of analytical grade, and were purchased from Sigma-Aldrich (Steinheim, Germany). Deionized water (Milli-Q) was used for the preparation of all solvents.

### 3.5. Polyphenol Analysis

#### 3.5.1. Apparatus and Chemicals

A Soxhlet apparatus was used for drug extraction. The quantitative analysis of the polyphenolic substances was performed with an Agilent 8453 UV/Vis spectrophotometer (Agilent, Germany) equipped with PC -HP 845x UV—Visible System (Agilent, Germany) and 1 cm quartz cuvettes.

With the exception of the Folin–Ciocalteu phenol reagent (FCR), casein (Merck, Darmstadt, Germany) and quercetin (Roth, Karlsruhe, Germany) all chemicals and reagents for polyphenol analysis were of analytical quality grade and were supplied by Kemika (Zagreb, Croatia).

#### 3.5.2. Total Polyphenol and Tannin Analysis (Folin–Ciocalteu Phenol Reagent (FCR) Procedure)

Total polyphenol (TP) and tannin (T) analysis was based on a reaction with FCR and spectrophotometric determination of TP and T at 720 nm-indirect analysis after precipitation T with casein [46,47]. The plant material (0.250 g; above ground parts) was previously extracted with methanol (30%, v/v), using water bath (70 °C) for 15 min [49,50]. The contents of TP and T in *V. saturejoides* extracts were evaluated in three independent analyses and were expressed as mg/g of dry weight of herbal material [48,49] according to equation:A = 1.069cexp. − 0.0029(1)
where A is absorbance and cexp. is measured concentration (µg/mL).

Total polyphenol and tannin analysis (FCR procedure) were based on a reaction with FCR and spectrophotometric determination of TP and T (indirect analysis after precipitation with casein) at 720 nm. Tannin was used as the standard substance.

#### 3.5.3. Total Flavonoid (TF) Analysis (TF Procedure)

TF included hydrolysis of glycosides (extraction of 0.20 g of powdered plant material with acetone, 25% HCl and 0.5% hexamethylenetetramine, using boiling water bath for 30 min), then extraction of total flavonoid aglycones with ethyl acetate, and complexation with AlCl_3_ [51,52]. Absorbance of the yellow complex was measured at 425 nm and concentration was calculated as quercetin using the following equation:% = A × 0.772/b; [A = absorbance; b = mass of dry plant material (g)](2)
where b is mass of the dry plant material (g) and 0.772 represents the conversion factor related to the specific absorbance of quercetin at 425 nm (i.e., 810). TF concentration was measured in three independent analyses and expressed as mg/g of dry weight of herbal material.

#### 3.5.4. Determination of Total Phenolic Acids (TPA) (TPA Procedure)

TPA procedure was performed according to official pharmacopoeial method [52] for determination of hydroxycinnamic derivatives. TPA were determined spectrophotometrically (three independent analyses) in extracts of *V. saturejoides* samples (0.200 g of powdered drug; ethanol 50% v/v; boiled water bath under a reflux condenser; 30 min), using the nitrite-molybdate reagent of Arnow, in a sodium hydroxide medium. 

TPA content, expressed as rosmarinic acid (λ = 505 nm), was calculated from the equation:A × 2.5/m; [A = absorbance; m = mass of the substance to be examined (g)](3)
where m is mass of the dry plant material (g) and 2.5 represents the conversion factor related to the specific absorbance of rosmarinic acid at 505 nm (i.e., 400).

TPA content, expressed as chlorogenic acid (λ = 525 nm), was calculated from the equation:A × 5.3/m; [A = absorbance; m = mass of the substance to be examined (g)](4)
where 5.3 represents the conversion factor related to the specific absorbance of chlorogenic acid at 525 nm (i.e., 188).

Conversion factors refer to the preparation of plant samples for spectrophotometric determination of flavonoid and phenolic acid contents according to official pharmacopoeial methods, considering the specific absorbances of standards.

### 3.6. Antioxidant Activity of Essential Oils and Hydrosols

#### 3.6.1. Oxygen Radical Absorbance Capacity Assay (ORAC)

The assay was performed in a Perkin–Elmer LS55 spectrofluorimeter, using 96-well white polystyrene microtiter plates (Porvair Sciences, Leatherhead, UK) according to a method described by Fredotovic et al. [53], with some adjustments due to different extracts. We had hydrophilic assay for hydrosols and lipophilic assay for EOs. For the hydrophilic assay, each reaction contained 180 µL of fluorescein (1 µM), 70 µL 2,2’-Azobis(2-methyl-propionamidine) dihydrochloride (AAPH, Acros Organics) (300 mM), and 30 µL of plant extracts or reference standard Trolox (6.25–50 µM) (Sigma–Aldrich). For the antioxidant test for hydrosols, all experimental solutions and samples were prepared in a phosphate buffer (0.075 mM, pH 7.0). We used absolute hydrosol and dissolved it in phosphate buffer 20× and 40× for the experiments. For the lipophilic assay, EO were dissolved in acetone (10 mg in 1 mL acetone). The EO acetone dilutions were further dissolved 40× and 80× in the phosphate buffer for the experiments. The measurements were performed in triplicate by a method described in Fredotovic et al. [53]. The ORAC values of hydrosols are expressed as μmol of Trolox equivalents (TE) per g of the total (undiluted) tested hydrosol sample. The ORAC values of EOs were expressed as μmol of Trolox equivalents (TE) per g of essential oil. The results were obtained from three independent experiments.

#### 3.6.2. Measurement of the DPPH Radical Scavenging Activity

The antioxidant capacity of the extracts was assessed by the DPPH method previously described by Mensor et al. and Payet et al. [54,55]. This method is based on the reduction of alcoholic DPPH (2,2-diphenyl-1-picrylhydrazyl) solution (Sigma–Aldrich) in the presence of a hydrogen-donating antioxidant using 96-well microtiter plates. Plant extracts as described in the ORAC method were used (acetone-dissolved essential oils and absolute hydrosols). We pipetted 100 µL methanol (Kemika, Zagreb, Croatia) and 200 µL standard and/or sample into each well. We prepared serial dilutions of standard and samples by pipetting 100 µL from the first row with a multichannel pipette into the wells in the second row and so on to the last row, where 100 µL of the solution is ejected after mixing. In the first column, in 96-well plates, a blank sample was always added. For EOs, the acetone and methanolic solution were used as blank and for hydrosols, water and methanolic solution were used as blank. The reaction starts by adding 100 µL of a methanolic solution of DPPH (200 µM) to each well. The initial absorbance at 517 nm was measured immediately, using MetOH as blank value. After 60 min incubation, the absorbance was measured again and the percentage of DPPH inhibition was calculated according to the following formula by Yen and Duh [56]:% inhibition = ((AC(0) − AA(t))/AC(0)) × 100,
where AC(0) is the absorbance of the control at t = 0 min, and AA(t) is the absorbance of the antioxidant at t = 1 h. All measurements were performed in triplicate. The standard curve was generated by plotting the percentage of inhibition of standard with corresponding μmol of Trolox. From the standard curve, results for EOs were expressed as μmol of Trolox per g of EO and for hydrosols as μmol of Trolox per g of absolute hydrosol. Because of the data from other relevant literature we also expressed IC50 values for EOs expressed in mg/mL.

For both antioxidant methods, we also tested the activity of the most abundant compound in EOs using the same method as for total oils. We used pure standard of the hexahydropharnesyl acetone (BOC Sciences, Shirley, NY, USA ), the concentration of the solution was 1 mg per g of acetone and was then further diluted in phosphate buffer up to the concentration of 100 μg/g.

### 3.7. Statistical Analysis

Statistical analysis was performed in GraphPad Prism Version 9. All data are expressed as mean ± SEM (n ≥ 3). The statistical significance for free volatile compounds, total phenolic compounds and antioxidant activity was assessed by multiple *t*-test, *p* < 0.05. Statistical tests were performed separately for lipophilic (essential oils) and hydrophilic fractions (hydrosols).

## 4. Conclusions

*Veronica* (family Plantaginaceae) is a very large and versatile plant genus, rich in biologically active specialized metabolites. In this research, free volatile compounds and their antioxidant activity were studied for the first time in *V. saturejoides* ssp. *saturejoides* from the two localities. The main volatile compounds in essential oils were hexahydrofarnesyl acetone and hexadecanoic acid. The main volatile compound in hydrosols was trans-1(7),8-p-mentadien-2-ol. For the genus *Veronic*a, the most studied specialized metabolites are iridoid glycosides, because of their importance in phylogeny and phenolic compounds, which have a great antioxidant activity. Our research showed that essential oils exhibit stronger antioxidant activity than hydrosols. Comparing the results from phenolic compounds in dry material of other investigated *Veronica* species with the results of phenolic and volatile compounds of this *Veronica* species, we can conclude that essential oils and hydrosols are valuable sources of potentially biologically active compounds. Free volatile compounds of the genus *Veronica* are only just being studied and we believe that such compounds have great potential as antioxidants in pharmacy and food technology.

## Figures and Tables

**Figure 1 plants-09-01646-f001:**
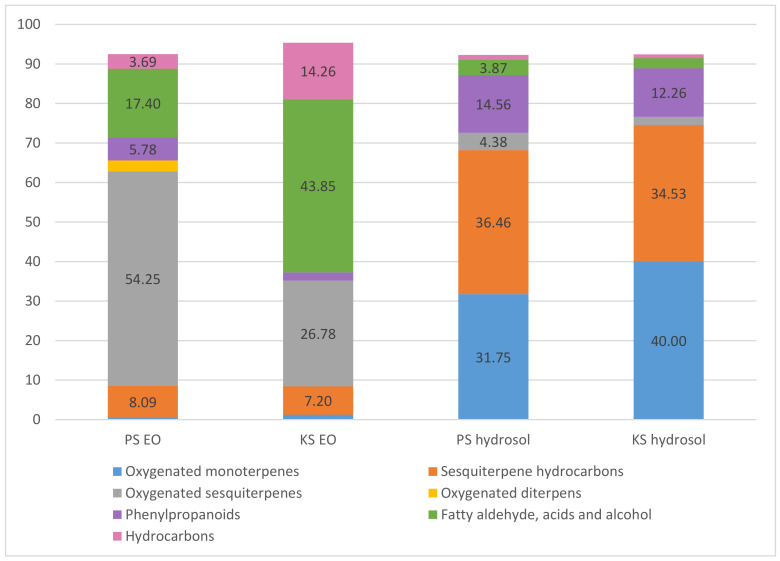
Distribution of volatile compounds by category in essential oils (%) and hydrosols (%); PS-Prenj sample, KS-Kamešnica sample.

**Figure 2 plants-09-01646-f002:**
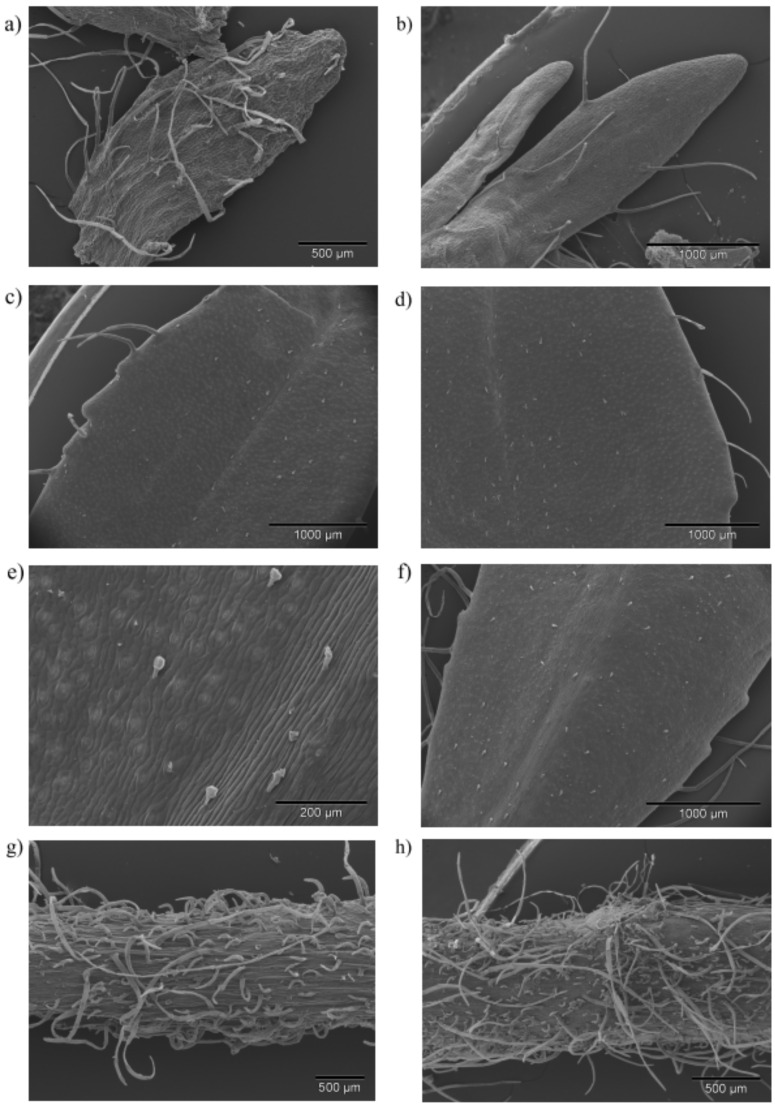
SEM micrographs of the different trichome types of *V. saturejoides* ssp. *saturejoides*. Rare (**a**,**b**) and dense (**g**,**h**) indumentum of non-glandular trichomes on the calyx (**a**,**b**) and stem (**g**,**h**); glandular trichomes on the adaxial (**c**,**d**) and abaxial (**e**,**f**) leaf surface; micrographs of Prenj (**a**,**c**,**e**,**f**,**g**) and Kamešnica (**b**,**d**,**h**) sample.

**Table 1 plants-09-01646-t001:** Chemical composition of the essential oil (%±SD) and hydrosols (%±SD) from the two samples from aerial parts of *V. saturejoides* Vis. ssp*. saturejoides.*

			Essential Oil Samples (%)		Hydrosol Samples (%)	
Component	RI ^a^	RI ^b^	PS	KS	PS	KS
***Oxygenated monoterpenes***			**0.44**	**1.23**	**31.75**	**40**
*trans-p-*Mentha*-*1(7),8-dien-2-ol	1187	1803	-	1.23 ± 0.01	31.75 ± 0.01 ^b^	36.63 ± 0.01 ^a^
Verbenone	1204	1705	-	-	-	0.76 ± 0.01
*endo*-Fenchyl acetate	1218	-	-	-	-	2.61 ± 0.01
Piperitone oxide	1365	-	0.44 ± 0.01	-	-	-
***Sesquiterpene hydrocarbons***			**8.09**	**7.2**	**36.46**	**34.53**
(*E*)***-***Caryophyllene *	1424	1585	0.94 ± 0.01 ^b^	2.46 ± 0.01 ^a^	24.52 ± 0.01 ^a^	12.35 ± 0.01 ^b^
*Z-*Methyl isoeugenol	1451	-	-	-	1.25 ± 0.01 ^b^	4.16 ± 0.01 ^a^
*allo-*Aromadendrene	1465	1662	2.35 ± 0.01 ^a^	0.68 ± 0.01 ^b^	8.13 ± 0.01 ^b^	11.53 ± 0.01 ^a^
*β-*Chamigrene	1476	1724	-	0.44 ± 0.01	-	1.17 ± 0.01
γ-Muurolene	1478	1685	0.39 ± 0.01 ^b^	0.87 ± 0.01 ^a^	-	0.65 ± 0.03
Germacrene D	1482	1692	2.78 ± 0.01 ^a^	1.77 ± 0.01 ^b^	2.56 ± 0.01 ^b^	4.67 ± 0.01 ^a^
δ-Cadinene	1517	1745	1.63 ± 0.01 ^a^	0.98 ± 0.01 ^b^	-	-
***Oxygenated sesquiterpenes***			**54.25**	**26.78**	**4.38**	**2.1**
Spathulenol	1577	2101	0.22 ± 0.01 ^b^	0.87 ± 0.01 ^a^	-	-
Caryophyllene oxide *	1581	1955	20.25 ± 0.01 ^a^	2.34 ± 0.01 ^b^	0.26 ± 0.01 ^b^	0.84 ± 0.05 ^a^
γ-Eudesmol	1632	2175	0.74 ± 0.01 ^a^	0.33 ± 0.01 ^b^	0.28 ± 0.01	-
α-Muurolol	1645	2181	2.91 ± 0.01	-	-	-
α-Bisabolol	1685	2210	-	-	0.32 ± 0.05	-
Hexahydrofarnesyl acetone	1839	2113	30.13 ± 0.01 ^a^	23.24 ± 0.01^b^	3.52 ± 0.01 ^b^	1.26 ± 0.01 ^a^
***Oxygenated diterpens***			**2.82**	-	*-*	*-*
Phytol	1942	2610	2.82 ± 0.03	-	-	-
***Phenylpropanoids***			**5.78**	**2.06**	**14.56**	**12.26**
Benzaldehyde	964	1513	-	-	0.75 ± 0.01 ^a^	0.34 ± 0.01 ^b^
2-Methoxy-4-vinylphenol	1294	2178	2.56 ± 0.01	-	0.46 ± 0.01	-
Methyl eugenol	1403	2005	1.28 ± 0.01 ^a^	1.13 ± 0.01^b^	13.35 ± 0.01 ^a^	11.92 ± 0.01 ^b^
Benzyl benzoate	1754	-	1.94 ± 0.01 ^a^	0.93 ± 0.01^b^	-	-
***Fatty aldehyde, acids and alcohol***			**17.4**	**43.85**	**3.87**	**2.59**
*n-*Nonanal	1100	1389	-	-	0.16 ± 0.01	-
Hexyl 2-methyl butanoate	1233	1425	-	-	1.13 ± 0.01 ^b^	1.82 ± 0.01 ^a^
Nonanoic acid	1267	2149	0.12 ± 0.01	-	-	-
Dodecanoic acid	1564	2480	0.59 ± 0.01 ^b^	1.67 ± 0.01 ^a^	-	-
1-Hexadecanol	1874	2371	7.73 ± 0.01 ^a^	4.53 ± 0.01 ^b^	1.32 ± 0.01	-
Hexadecanoic acid	1959	2912	7.88 ± 0.01 ^b^	37.31 ± 0.01 ^a^	1.26 ± 0.01 ^a^	0.77 ± 0.01 ^b^
Oleic acid	2133	-	1.08 ± 0.01 ^a^	0.34 ± 0.03 ^b^	-	-
***Hydrocarbons***			**3.69**	**14.26**	**1.27**	**0.95**
Heneicosane *	2100	2100	0.42 ± 0.02 ^b^	0.68 ± 0.07 ^a^	-	-
Docosane *	2200	2200	0.73 ± 0.01 ^b^	3.27 ± 0.01 ^a^	1.27 ± 0.01 ^a^	0.95 ± 0.01 ^b^
Tricosane *	2300	2300	1.27 ± 0.01 ^b^	4.03 ± 0.01 ^a^	-	-
Tetracosane *	2400	2400	0.59 ± 0.05	-	-	-
Pentacosane *	2500	2500	0.68 ± 0.01 ^b^	6.28 ± 0.01 ^a^	-	-
***Total (%)***			**92.47**	**95.38**	**92.29**	**92.43**

RI^a^, retention indices on capillary column VF5-ms; RI^b^, retention indices on capillary column CP-Wax 52; RI, identification by comparison to literature [25], and/or homemade library, comparison of mass spectra with those in mass spectral libraries NIST02 and Wiley 9; * co-injection with reference compounds; –, not identified; SD = standard deviation of triplicate analysis; significant differences were determined using multiple *t*-test. ^a^,^b^—Mean values with different superscript letters indicate a statistically significant difference between data from two locations (*p* < 0.05); PS—Prenj sample, KS—Kamešnica sample.

**Table 2 plants-09-01646-t002:** Occurrence and frequency of trichomes on aerial parts of *Veronica saturejoides* ssp. *saturejoides*.

Sample	Trichome	Leaf		Calyx	Stem
	Type	Adaxial	Abaxial		
Prenj	Attenuate *	±	±	+	++
	capitate C1	±/+	±/+	–/±	±/+
Kamešnica	attenuate	±	±	+	+/++
	capitate C1	±/+	±/+	–/±	±

Note: * attenuate, non-glandular trichomes; trichomes: – absent, ± rare, + present, ++ abundant.

**Table 3 plants-09-01646-t003:** Contents of total polyphenols (TP), total tannins (T), total flavonoids (TF), and total phenolic acids (TPA) in *V. saturejoides*.

Species	TP (mg/g DW)	T (mg/g DW)	TF (mg/g DW)	TPA (505 nm) (mg/g DW)	TPA (525 nm) (mg/g DW)
*V. saturejoides* (KS)	86.9 ±1.4 ^a^	2.3 ± 1.3	0.8 ± 0.00	33.1 ± 1.7 ^a^	65.6 ± 0.2 ^a^
*V. saturejoides* (PS)	70.9 ± 0.9 ^b^	1.7 ± 0.5	0.8 ± 0.00	19.5 ± 0.2 ^b^	45.4 ± 2.1 ^b^

Note: DW, dry weight; SD = standard deviation of triplicate analysis; significant differences were determined using multiple *t*-test. ^a^,^b^—Mean values with different superscript letters indicate a statistically significant difference between data from two locations (*p* < 0.05); KS—Kamešnica sample, PS—Prenj sample.

**Table 4 plants-09-01646-t004:** Polyphenolic analyses of hydrosols.

Samples	Vanillin	Cinnamic Acid	Protocatechuic Acid
*V. saturejoides* (KS)	0.22 ± 0.01	0.12 ± 0.02	7.33 ± 0.35
*V. saturejoides* (PS)	-	-	-

Results expressed in mg/L of hydrosol sample; KS—Kamešnica sample, PS—Prenj sample.

**Table 5 plants-09-01646-t005:** Antioxidant potential of *V. saturejoides* essential oil and hydrosol determined by ORAC and DPPH method.

	Essential Oil		Hydrosols		Hexahydropharnesyl Acetone
Antioxidant Assay	PS	KS	PS	KS	
ORAC (Trolox eq)	255.1 ± 2.54	256.5 ± 5.73	0.559 ± 0.059	0.679 ± 0.036	-
DPPH (Trolox eq)	20.73 ± 0.21 ^b^	44.32 ± 0.13 ^a^	0.225 ± 0.062	0.323 ± 0.014	-
DPPH (% inhibition)	46.23 ± 4.37 ^b^	66.99 ± 2.98 ^a^	34.84 ± 0.89 ^b^	49.26 ± 1.7 ^a^	-
DPPH (IC 50)	10.88 ± 1.24 ^b^	7.16 ± 0.12 ^a^	-	-	-

ORAC, oxygen radical absorbance capacity, results for EOs expressed as μmol of Trolox equivalents (TE) per g of EO (10 mg/mL) and for hydrosols as μmol of Trolox equivalents (TE) per g of the total (undiluted) tested hydrosol sample; DPPH, results for EOs expressed as μmol of Trolox per g of EO (10 mg/mL) and for hydrosols as μmol of Trolox per g of absolute hydrosol, IC50 expressed in mg/mL for EOs; –showed no activity; SD = standard deviation of triplicate analysis; significant differences were determined using multiple t-test. ^a^,^b^—Mean values with different superscript letters indicate a statistically significant difference between data from two locations (*p* < 0.05) KS—Kamešnica sample, PS—Prenj sample.

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
