# Peer review of "Endemic Veronica saturejoides Vis. ssp. saturejoides–Chemical Composition and Antioxidant Activity of Free Volatile Compounds"

_plants, 2020, doi:10.3390/plants9121646_

Round 1
Reviewer 1 Report
The paper entitled ‘Endemic Veronica saturejoides Vis. ssp. saturejoides – Chemical Composition and Antioxidant Activity of Free Volatile Compounds’ provides interesting results about essential oils and other selected secondary metabolites in Veronica plants from two separate sites. The presented data fully covers a spectrum of Plants interests. However, before acceptation I have some comments, which are highlighted in the attached copy.

Author Response
Dear Reviewer,
Thank you for reviewing the manuscript "Endemic Veronica saturejoides Vis. ssp. saturejoides – Chemical Composition and Antioxidant Activity of Free Volatile Compounds" and for your comments and suggestions. We have added our answers in the attached PDF file where your comments were. We cannot upload two documents at once so we couldn't attach Word document with Track changes made.
Best regards,
Authors

Reviewer 2 Report
Dear Authors:
The manuscript is well-written and very comprehensive. I do not have any other comments or suggestions.
Author Response
Dear Reviewer,
Thank you for reviewing the manuscript : “ Endemic Veronica saturejoides Vis. ssp. saturejoides – chemical composition and antioxidant activity of free volatile compounds” and for all your nice comments.
Best regards,
Authors

Reviewer 3 Report
In the manuscript titled: “ Endemic Veronica saturejoides Vis. ssp. saturejoides – chemical composition and antioxidant activity of free volatile compounds” Authors investigated chemical content and antioxidant activity of the essential oils and hydrosols from aerial parts of V. saturejoides samples.
Investigations consisted of GC-MS for qualitative analysis of essential oil and hydrosols constituents, HPLC for peholic compounds in hydrosols assessment and spectrophotometric antioxidant activity determinations. Additionally Authors have made micromorphological investigations to find site of essential oil production.
The experiment has been correctly planned, but in my opinion some of the methodological parts are very complicated. For example method of GC-MS analysis of hydrosols samples (L: 326-335). This entails the risk of losing the repeatability of the results. In this section, the authors should provide information on the number of repeated extractions and the distribution of the obtained results.
Please explain the conversion factors used in more detail. What do the values in parentheses (i.e. 810) mean for quercetin (L: 411), (i.e. 400) for rosmarinic acid (L: 425) and (i.e. 188) for chlorogenic acid (L: 429).
Other comments:
Table 1 : There are no concentration units
L: 128-129: There is repetition of the same sentence
Table 4 and Table 5 could be combined
L: 385-387 and L: 392-393: : There is repetition of the same sentence
Author Response
Dear Reviewer,
Thank you for reviewing the manuscript : “ Endemic Veronica saturejoides Vis. ssp. saturejoides – chemical composition and antioxidant activity of free volatile compounds” and for all your constructive comments.
The following changes were done:
Reviewer #2: The experiment has been correctly planned, but in my opinion some of the methodological parts are very complicated. For example method of GC-MS analysis of hydrosols samples (L: 326-335). This entails the risk of losing the repeatability of the results. In this section, the authors should provide information on the number of repeated extractions and the distribution of the obtained results.
Authors answer: All the results were obtained from three independent analyses, this wasn't mentioned in the Materials and Methods section so we added this. Line 387-389.
Reviewer #2: Please explain the conversion factors used in more detail. What do the values in parentheses (i.e. 810) mean for quercetin (L: 411), (i.e. 400) for rosmarinic acid (L: 425) and (i.e. 188) for chlorogenic acid (L: 429).
Authors answer: Conversion factors refer to the preparation of plant samples for spectrophotometric determination of flavonoid and phenolic acid contents according to official pharmacopoeial methods, taking into account the specific absorbances of standards.
The expression representing the specific absorbance of a dissolved substance refers to the absorbance of a 10 g/L solution in a 1 cm cell and measured at a defined wavelength.
e = molar absorptivity
Specific absorbances for used standards: quercetin at 425 nm ( = 810), rosmarinic acid at 505 nm ( = 400), and chlorogenic acid at 525 nm ( = 180).
We have added part about this in Materials and Methods. Line 497-499
Other comments:
Reviewer #2: Table 1: There are no concentration units
Authors answer: Line 180. Added concentration units (percentage %) in the Table description and in the Table and in the Materials and Methods section – Line 388.
Reviewer #2: L: 128-129: There is repetition of the same sentence
Authors answer: Deleted the repeated sentence. Line 155.
Reviewer #2: Table 4 and Table 5 could be combined
Authors answer: Tables 4 and 5 were combined in Table 5. Also, we have added an additional row to show DPPH activity in percentage (%) of inhibition for other authors to have more options for comparing their results. Also, when doing a statistical interpretation of the results we have detected our mistake in expressing units of measurement for DPPH activity (Trolox eq) so we have corrected this in the table 5 and in the Table legend. Line 312
Reviewer #2: L: 385-387 and L: 392-393: There is repetition of the same sentence
Authors answer: Line 449-450. Deleted repeated sentence.
Best regards,
Authors

Reviewer 4 Report
The manuscript presents interesting data on the phytochemical composition of hydrosol and essential oil obtained from Veronica saturejoides Vis. ssp. Saturejoides, species endemic to Croatia, Bosnia and Herzegovina, and Montenegro. The content of various groups of plant metabolites was described with the combination of chromatographic and chemical methods. This ascertains its novelty. The analytical methods are well described and developed, and this work seems interesting and important for the future usage of the analyzed fraction for the food, cosmetic and pharmacological industry. What is more, the antioxidant activity of of essential oils and hydrosols was assessed. This is the first study of the Veronica saturejoides, which ascertains its novelty.
General remarks
1. Present work properly describes the various aspects of phytochemical analysis, engaging chromatographically and chemical detection methods, as well as biological tests, in order to provide a broad insight into the phytochemistry of the V.saturejoides. 2. The manuscripts need some corrections in order to avoid inconsistency and impose the fluency of argumentation – as listen in Minor remarks section.3. The linguistic competitions of the reviews are not on an appropriate level to justify the linguistic correctness of the manuscript, but In the modest and opinion, the text would be more fluent with some connectors such, as therefore, moreover, furthermore, in contrast, etc. which gives the text logical structure and fluency. In my modest opinion, the different parts of the manuscript differ significantly in terms of the language – the part of the results section 2.2. is much more fluent and coherent then the previous part of the manuscript considering phytochemical analysis.
Specific remarks
Abstract:
In the opinion of the Reviewr it would be better to group information on hydrosol composition – i.e. sentence “ the line 25-27 could be followed by the sentence “Group of oxygentated ..”
Lines 32-33 It would be more justified to write That “Obtained results demonstrate that this genus is potential source of volatiles with antioxidant activity” The volatile are not a source of antioxidants, but in the matter of fact, exhibit antioxidant activity.
Lines 33-34 The last sentence is separated from the prior description of the phytochemical identification (line 25-29), and could be grouped with it to avoid any confusions.
Introduction:
The introduction should be divided into paragraphs to rise the legibility of the manuscript. The paragraphs could begin from the beginning of the sentences present in the line 56, 65, 78.
Line 77 – not only in pharmacy or agriculture – Hydrosols from various plants are becoming increasingly important in the cosmetic industry, as a part of complex recipes as well as independent products, and this application could be mentioned in the Introduction for the extending of the explanation of the study importance and potent application.
Line 81 – it is justified to mention that these metabolites are produced also in response to biological stress such as pathogen infection.
Lines 83-86 These sentence requires reorganization to give it more logical structure, as the tree parts of the sentence aren’t coherent. The Reviewer would suggest to rearrange in the following way, what is only an illustration of the problem: These synthetic compounds can also be carcinogenic when used in canned foods, therefore the search for safe natural food preservatives goes on and this research contributes to the study of natural products as potential antioxidants in food preservations. What is more, it is basic research on the bioactive compounds of Veronica genus.
Results and discussion:
Lines 93, 95 – the abbreviation PS and KS should be explained when they appear in the text for the first time, as well as in Tables and Figures captions.
Lines 104, line 107 – the sentences could be reconstructed in order to specify what is the result of the presented study and what is the discussion, for example by dividing the sentence and by putting stress on the new results.
Lines 112-114 – repetition of the sentence.
Line 128-129 - repetition of the sentence.
Lines 132-136 – the comparison of the results obtained for other species requires some comments, not only listing of the obtained compounds.
Line 138 –I would suggest to begin a new paragraph here, as we shift from the comparison of different species to the description of the different groups of volatile compounds in the investigated species.
Line 209 – change “identify” for “detected”, as identification is most of the connected with the analysis of detected compounds.
Material and Methods:
In some places, the names of the producers of reagents should be inserted.
Author Response
Dear Reviewer,
Thank you for all your comments for the manuscript „Endemic Veronica saturejoides Vis. ssp. saturejoides – Chemical Composition and Antioxidant Activity of Free Volatile Compounds”.
The following changes were done:
English corrections: The linguistic competitions of the reviews are not on an appropriate level to justify the linguistic correctness of the manuscript, but In the modest and opinion, the text would be more fluent with some connectors such, as therefore, moreover, furthermore, in contrast, etc. which gives the text logical structure and fluency. In my modest opinion, the different parts of the manuscript differ significantly in terms of the language – the part of the results section 2.2. is much more fluent and coherent then the previous part of the manuscript considering phytochemical analysis.
Authors answer: Previous version of the manuscript “Selected specialized metabolites and micromorphological traits of the endemic Veronica saturejoides Vis. ssp. saturejoides” was professionally edited by MDPI's English editing service. For this new version of the manuscript English editing (new parts) was done through the application InstaText. We have additionally once more submitted the paragraph 2.2. through InstaText. Line 101-169. This part was also addapted as other reviewer asked so it is rearranged.
Abstract:
Reviewer #3: - In the opinion of the Reviewer it would be better to group information on hydrosol composition – i.e. sentence “ the line 25-27 could be followed by the sentence “Group of oxygentated ..”
Authors answer: We accepted the reviewer's proposal and grouped the data on the composition of the essential oil and hydrosol. Line 27-28
Reviewer #3: Lines 32-33 It would be more justified to write That “Obtained results demonstrate that this genus is potential source of volatiles with antioxidant activity” The volatile are not a source of antioxidants, but in the matter of fact, exhibit antioxidant activity.
Authors answer: Changed according to Reviewer’ suggestion. Line 36-37
Reviewer #3: Lines 33-34 The last sentence is separated from the prior description of the phytochemical identification (line 25-29), and could be grouped with it to avoid any confusions.
Authors answer: Changed according to Reviewer’ suggestion. Line 23-25
Introduction:
Reviewer #3: The introduction should be divided into paragraphs to rise the legibility of the manuscript. The paragraphs could begin from the beginning of the sentences present in the line 56, 65, 78.
Authors answer: Changed according Reviewer’ suggestion. Divided into paragraphs. Line 56 to 58, line 65 to 68, line 78 to 85.
Reviewer #3: Line 77 – not only in pharmacy or agriculture – Hydrosols from various plants are becoming increasingly important in the cosmetic industry, as a part of complex recipes as well as independent products, and this application could be mentioned in the Introduction for the extending of the explanation of the study importance and potent application.
Authors answer: Changed according to Reviewer’ suggestion. We have added the sentence Line 80-84. Also, reference for this sentence was added in the literature – line 546, so because of this all numbers for the references from number 13-54 were changed in text and in References.
Reviewer #3: Line 81 – it is justified to mention that these metabolites are produced also in response to biological stress such as pathogen infection.
Authors answer: Changed according Reviewer’ suggestion. Line 88
Reviewer #3: Lines 83-86. These sentence requires reorganization to give it more logical structure, as the tree parts of the sentence aren’t coherent. The Reviewer would suggest to rearrange in the following way, what is only an illustration of the problem: These synthetic compounds can also be carcinogenic when used in canned foods, therefore the search for safe natural food preservatives goes on and this research contributes to the study of natural products as potential antioxidants in food preservations. What is more, it is basic research on the bioactive compounds of Veronica genus.
Authors answer: Changed according Reviewer’ suggestion. Line 91-94, added Reviewers suggestion to the end of Paragraph.
Results and discussion:
Reviewer #3: Lines 93, 95 – the abbreviation PS and KS should be explained when they appear in the text for the first time, as well as in Tables and Figures captions.
Authors answer: Changed according Reviewer’ suggestion. Line 101 added explanation for PS and Line 102 added explanation for KS, Line 186 added explanation for PS and KS under the Table 1. Line 188-189, added explanation for PS and KS under Figure 1. Line 252 – added explanation for PS and KS under the Table 3. Line 277 - added explanation for PS and KS under the Table 4. Line 319 - added explanation for PS and KS under the Table 5.
Reviewer #3: Lines 104, line 107 – the sentences could be reconstructed in order to specify what is the result of the presented study and what is the discussion, for example by dividing the sentence and by putting stress on the new results.
Authors answer: Changed according Reviewer’ suggestion. Lines 128-138 – Sentences were reconstructed so that first part of the paragraph are results and other half represents disscussion.
Reviewer #3: Lines 112-114 – repetition of the sentence.
Authors answer: Deleted the repeated sentence. Line 139-140.
Reviewer #3: Line 128-129 - repetition of the sentence.
Authors answer: Deleted the repeated sentence. Line 155.
Reviewer #3: Lines 132-136 – the comparison of the results obtained for other species requires some comments, not only listing of the obtained compounds.
Authors answer: Changed according Reviewer’ suggestion. Line 162-163. We added sentence about comparison of the results.
Reviewer #3: Line 138 –I would suggest to begin a new paragraph here, as we shift from the comparison of different species to the description of the different groups of volatile compounds in the investigated species.
Authors answer: This paragraph was transferred to the beginning of the Section 2.1. as requested by other reviewer. Line 100-109.
Reviewer #3: Line 209 – change “identify” for “detected”, as identification is most of the connected with the analysis of detected compounds.
Authors answer: Changed according Reviewer’ suggestion. Line 262, „identified“ changed to „detected“.
Material and Methods:
Reviewer #3: In some places, the names of the producers of reagents should be inserted.
Authors answer: Inserted producers of reagents that were missing according Reviewer’ suggestion. Line 355, 507-509, 525, 528, 547.
Kind regards,
Authors

Round 2
Reviewer 1 Report
Dear Authors,
thank you for explanations.
Best regards.